# Epidemiological distribution of *Echinococcus granulosus s.l.* infection in human and domestic animal hosts in European Mediterranean and Balkan countries: A systematic review

**Francesca Tamarozzi[1], Matteo Legnardi[2], Andrea Fittipaldo[1], Michele Drigo[2], Rudi Cassini[2]***

**1** Department of Infectious-Tropical Diseases and Microbiology, IRCCS Sacro Cuore Don Calabria Hospital, Negrar (Verona), Italy, **2** Department of Animal Medicine, Production and Health, University of Padova, Legnaro (Padova), Italy

* rudi.cassini@unipd.it

## Abstract

Cystic echinococcosis (CE) is a neglected zoonosis caused by infection with the cestode *Echinococcus granulosus sensu lato*. We carried out a systematic literature review on *E. granulosus s.l.* human and animal (cattle, sheep, dog) infection in European Mediterranean and Balkan countries in 2000–2019, to provide a picture of its recent epidemiology in this endemic area. MEDLINE, EMBASE, Scopus, Google Scholar and Open Grey databases were searched. Included cases were: i) for humans, data from hospital records and imaging studies; ii) for dogs, data from necropsy and coprological studies; iii) for ruminants, cases based on slaughter inspection. The NUTS (Nomenclature of Territorial Units for Statistics) classification was used to categorize extracted data in epidemiological units, defined as data referred to one NUTS2 (basic region) in one year time. Data were then aggregated to NUTS1 level (major regions), calculating the average incidence value of included epidemiological units. For prevalence studies covering different epidemiological units, the pooled prevalence was estimated. Data were extracted from 79 publications, 25 on human infection (covering 437 epidemiological units), and 54 on animal infection (52 epidemiological units for cattle, 35 for sheep and 25 for dogs). At NUTS1 level, average annual incidence rates of human CE ranged from 0.10–7.74/ 100,000; pooled prevalence values ranged from 0.003–64.09% in cattle, 0.004–68.73% in sheep, and 0–31.86% in dogs. Southern and insular Italy, central Spain, Romania and Bulgaria reported the highest values. Bovine data showed a more similar pattern to human data compared to sheep and dogs. Limitation of evidence included the paucity of human prevalence studies, data heterogeneity, and the patchy geographical coverage, with lack of data especially for the Balkans. Our results confirm Italy, Spain, and Eastern Europe being the most affected areas, but data are extremely heterogeneous, geographical coverage very patchy, and human prevalence studies extremely scant. Results also highlight the notorious problem of underreporting of *E. granulosus s.l.* infection in both humans and animals.

**Data Availability Statement:** All relevant data are within the manuscript and its Supporting Information files.

**Funding:** RC received a grant (BIRD174940), by the Department of Animal Medicine, Production and Health of the University of Padova (https://www.maps.unipd.it/). The funder had no role in study design, data collection and analysis, decision to publish, or preparation of the manuscript.

**Competing interests:** The authors have declared that no competing interests exist.

## Author summary

Cystic echinococcosis (CE) is a neglected zoonosis caused by infection with the parasite *Echinococcus granulosus sensu lato*, naturally transmitted between canids and livestock; CE in humans can be a serious condition. In endemic areas, CE is responsible for significant health and economic losses, but its real burden is difficult to estimate. *E. granulosus s. l.* is especially present in areas where livestock breeding is practiced, including European Mediterranean and Balkan countries. We carried out a systematic literature review on the epidemiology of *E. granulosus s.l.* human and animal infection in this area in 2000–2019. Data were extracted from 79 publications, and referred to Nomenclature of Territorial Units for Statistics (NUTS) levels per year. Average annual incidence rates of human CE ranged from 0.10–7.74/100,000; pooled prevalences ranged from 0.003–64.09% in cattle, 0.004–68.73% in sheep, and 0–31.86% in dogs. Bovine data showed a more similar pattern to human data compared to sheep and dogs. Our results confirm that Italy, Spain, and Eastern Europe are the most affected areas, but data are extremely heterogeneous, geographical coverage very patchy, and human prevalence studies extremely scant. Results also highlight the well-known problem of underreporting of *E. granulosus s.l.* infection in both humans and animals.

## Introduction

Cystic echinococcosis (CE) is a neglected parasitic zoonosis caused by infection with the cestode *Echinococcus granulosus sensu lato* species complex. The parasite is endemic worldwide, especially prevalent in areas where livestock breeding is practiced [1]. It is naturally transmitted between canids, definitive hosts harbouring the adult cestodes in the intestine and shedding parasite eggs with the faeces, and livestock, intermediate hosts getting infected upon ingestion of parasite eggs, where the larval stage (metacestode) develops in the form of fluid-filled cysts in liver, lungs and other organs. The definitive hosts in turn acquire the infection by eating parasite cysts in infected organs of slaughtered animals. Humans act as accidental, dead-end intermediate hosts, acquiring the infection through ingestion of parasite eggs and developing echinococcal cysts mostly in the liver, followed by lungs [2].

*E. granulosus s.l.* is particularly prevalent in China and Central Asia, South America, North and East Africa, and Australia; in Europe, another endemic area, it is especially present in the Mediterranean and Eastern countries, its cycle mainly involving domestic dogs and ruminants (sheep, goats, cattle) [1]. CE is responsible for significant economic losses in the public health sector. At global level, Budke and colleagues [3] estimated a human burden of around 1 million Disability-Adjusted Life Years (DALYs) and 760 million US$ losses due to human infection (accounting for underreporting), and annual livestock production losses of at least US$ 140 million. For what concerns Europe, in Italy, based on Hospital Discharge Records (HDR), Piseddu and colleagues [4] estimated a financial burden due to human CE of around € 53 million in 2001–2014, with a national average economic burden of € 4 million per year. In Spain, Benner and colleagues [5] estimated, for the year 2005, and overall economic loss due to human and livestock CE of about € 150 million, of which about 130 million were human-associated and about 16 million animal-associated.

However, the real health and economic burden due to CE are difficult to estimate. On the one hand, animal infection is not perceived as an infection of high concern, and presence and implementation of structured surveillance of infection in animals is variable among and within

countries [6]. On the other hand, human CE is a chronic, disabling condition, its clinical manifestations ranging from asymptomatic infection to extremely severe disease, and costs for its treatment may be substantial. However, again, presence and type of disease notification systems vary greatly from country to country, and mostly include only hospitalized cases [6]. This situation results in poor accuracy, and likely large underestimate, of the epidemiological parameters of CE distribution, costs, and socio-economic burden. This in turn contributes to the neglect of CE, with subsequent little focus on accurate and representative data collection, feeding a vicious circle of data inaccuracy, underestimation, and neglect.

Here, we carried out a systematic review of the literature on *E. granulosus s.l.* infection prevalence and incidence data in European Mediterranean and Balkan countries between 2000 and 2019, in both humans and animals, in an attempt to provide a picture of recent animal and human CE epidemiology in Europe, and their relation. This geographical area, where *E. granulosus s.l.* is endemic, was chosen on the basis of relative homogeneity of climate and livestock breeding practices.

## Materials and methods

### Search strategy

The research question of the study concerned the current epidemiological situation, in terms of incidence and/or prevalence of human, canine, and domestic ruminant hosts, in European Mediterranean and Balkan countries. A literature search was carried out to identify all possible studies that could help to answer the research question. The following databases were searched for relevant studies: MEDLINE (PubMed) (1966 to October 19th 2019) and EMBASE (1974 to October 19th 2019). The detailed strategy is available as Supplementary Information S1 Text. Additional sources were searched up to October 2019: Google Scholar, Scopus, and Open Grey were used to identify articles that cited relevant reports using free-text terms ("cystic echinococcosis", "hydatidosis", "Echinococcus granulosus"). For Google Scholar, Scopus, and Open Grey, the first 500 search results, sorted by date, were considered; this arbitrary number was chosen as it reasonably included all relevant results for the investigated time frame. The reference lists of reviews and relevant reports were also searched to identify additional studies. Search results were combined and duplicates removed before screening for relevance. No restriction was applied regarding language or publication status (published or in press). The search was initially performed on papers published from January 1st 1980, however, due to the difficulty in extracting data from literature published during '80 and '90, eventually only papers published from January 1st 2000 onwards were included in the analysis. Among these, retrospective studies reporting data from investigations starting before 1980 were also excluded. The work is presented according to the recommendations of the Preferred Reporting Items for Systematic Reviews and Meta-Analyses (PRISMA http://prisma-statement.org/prismastatement/Checklist.aspx, S1 PRISMA Check list).

### Population, inclusion and exclusion criteria, study design, and outcomes

Original cross-sectional and longitudinal studies, as well as case series, reporting prevalence or incidence of human or animal (ovine, bovine, and canine) *E. granulosus s.l.* infection were included in this review; case reports and review articles were excluded. Papers reporting data from the following European Mediterranean and Balkan countries were included (from West to East): Portugal, Spain, France, Italy, Slovenia, Serbia, Croatia, Bosnia-Herzegovina, Montenegro, Macedonia, FYROM (Former Yugoslav Republic Of Macedonia), Republic of North Macedonia, Kosovo, Albania, Greece, Romania, Bulgaria and Cyprus. No restriction was applied regarding publication type (e.g. research paper or conference report) or setting (e.g. field or clinical setting). For human studies, only cases reported from HDR or hospital

databases, or confirmed by histopathology, or based on imaging studies, were included; surveys based on serology only were excluded due to the poor performance of serology for CE in population studies [7]. For dog studies, necropsy and coprological studies based on microscopy, PCR or copro-ELISA were included. For ruminants, studies based on pathological examination after slaughter were included. Extracted and analyzed study outcomes were prevalence or incidence of *E. granulosus s.l.* infection based on reported incidence or, when available, on reported cases (numerator) and size of the at risk population (denominator).

## Study selection and data extraction

Two authors (FT and ML for human studies and ML and MD for animal studies) reviewed titles and abstracts of publications identified by the search, in order to identify all studies that potentially met the inclusion criteria. After having obtained the full text, the two authors independently assessed whether the study was eligible for inclusion. Potentially eligible studies were excluded if: 1) full text and abstract were both unavailable or only the abstract was available but did not convey the needed data; 2) infection cases (numerator) and size of the studied population (denominator) or calculated incidence/prevalence were not extractable; 3) diagnostic method was not mentioned or not eligible; 4) *E. granulosus s.l.* was not investigated or the species of *Echinococcus* was not specified; 5) study duplication. The same two authors performed the data extraction using a pre-designed Excel data extraction sheet. At all steps, when the two authors disagreed and did not reach a consensus after discussion, a third author (RC) facilitated the discussion and eventually made the final decision. The following data were extracted: country and geographical area investigated, at NUTS1 (Nomenclature of Territorial Units for Statistics) and NUTS2 level [8], date and type of study (cross-sectional, prospective or retrospective case cohorts and case series), host species investigated, type of epidemiological parameter (prevalence or incidence) measured, diagnostic method, case numbers (numerator) and studied population (denominator) or calculated incidence, and type of studied population (general or specific setting/subpopulation). The decision to refer extracted data to the NUTS level was due to the need of clustering data in regions of comparable size, at least in terms of human population. The NUTS classification is a hierarchical system for dividing the territory of the European countries. Each country is composed by one or more major regions (NUTS1 level) and each NUTS1 is divided in different NUTS2 areas. According to NUTS regulation, minimum and maximum population thresholds for the size of NUTS1 and NUTS2 levels are 3,000,000–7,000,000 and 800,000–3,000,000, respectively. The study area consisted of 46 NUTS1 and 110 NUTS2 (French and Portuguese overseas territories were not considered).

## Data analysis and synthesis

The human CE dataset (S1 Table) was organized by extracting the data from all eligible papers at the level of NUTS2 for what concerns the spatial aspect and considering the timeframe of one year. This was considered as the epidemiological unit. Data from publications investigating different years and/or different NUTS2 were divided in as many epidemiological units as appropriate. Therefore each row of the dataset reported the data of a single epidemiological unit (i.e. NUTS2 level in a given year), in terms of prevalence, incidence, number of cases (numerator), and/or total population at risk (denominator).

For human case series, when both incidence rate and number of cases were reported, the population at risk was calculated. If data reported in multi-year retrospective case series of human CE were not differentiated for each year, a single value was inserted in the dataset, corresponding to an average annual incidence. If the NUTS2 level was not specified, data were referred to the NUTS1 level, and if also the NUTS1 specification was absent, data were referred

to the national level. To estimate incidence at NUTS1 level, the average of all values reported for all epidemiological units belonging to the same NUTS1 was calculated.

In studies where human infection was investigated through a cross-sectional survey, the sum of positive cases (numerator) and tested individuals (denominator) was calculated for each epidemiological unit. To merge data from different epidemiological units at NUTS1 level, the pooled prevalence and 95% Confidence Interval (CI) were estimated, based on the inverse variance method and logit transformation [9] to account for study weight with respect to population size (i.e. sampling fraction) and variance effect (i.e. prevalence very low or very high).

Similarly, the animal infection dataset (S2 Table) was organized spatially at the level of NUTS2 and considering the timeframe of one year, keeping separately the different host species investigated (dog, cattle, sheep). The data analysis was carried out as described above, for prevalence estimation.

Data management and elaboration were performed in Excel 14.7.7 (Microsoft Corporation, Redmond, WA).

## Results

### Bibliographic search

The search and selection of included studies is shown in Fig 1. The database search effectuated on February 19th, 2018 retrieved a total of 811 publications. After duplicates were removed, 572 records were further filtered by year of publication, leaving 471 papers published on or after the year 2000. The literature search re-launched on October 28th, 2019 retrieved further 24 records; further 5 potentially eligible studies published on or after year 2000 were retrieved from the bibliography lists of reviews. A total of 500 records were therefore screened for potential eligibility by title and abstract, of which 103 were selected for full-text review: 37 on human infection, 8 on both human and animal infection, and 58 on animal infection. Of these, 24 were further excluded, leaving 79 publications from which data were extracted: 25 on human infection and 54 on animal infection. It was not possible to extract data for both human and animal infections from those studies investigating the two aspects at the same time.

### Human infection

Data extracted from the 25 publications on human infection were covering 7 eligible countries. All but two papers [10,11] were retrospective case series: hospitalized cases (n = 15) based on HDR, or based on clinical records of cases who reached clinical attention (n = 6), or surgical cases (n = 2). The above mentioned two papers were abdominal ultrasound-based cross-sectional surveys, conducted on the rural populations of three countries, among which two were eligible for our study (Romania and Bulgaria).

The 23 retrospective case series studies were mostly reporting multi-year data: 14 papers recorded incidence values for each year, in eight publications a single incidence value was reported for the whole period, and one study investigated a 1-year period. In seven papers only the incidence value was reported, in 15 both the number of new cases and the incidence values, while only in one paper the population at risk was also explicitly reported (around 650,000 people, population of Timis County in Romania–found in the NUTS1 RO4). Overall, extracted data were separated in 430 epidemiological units, distributed in 33 NUTS1. The average incidence for each NUTS1 is reported in Table 1, together with the lowest and highest values, if more than one epidemiological unit were present in the same NUTS1. In these cases, the type of data aggregation is specified, namely multi-years if different (usually subsequent) years of the same NUTS2 were included in the study, or multi-area if different NUTS2 were assessed. The spatial distribution of the incidence values is shown in Fig 2.

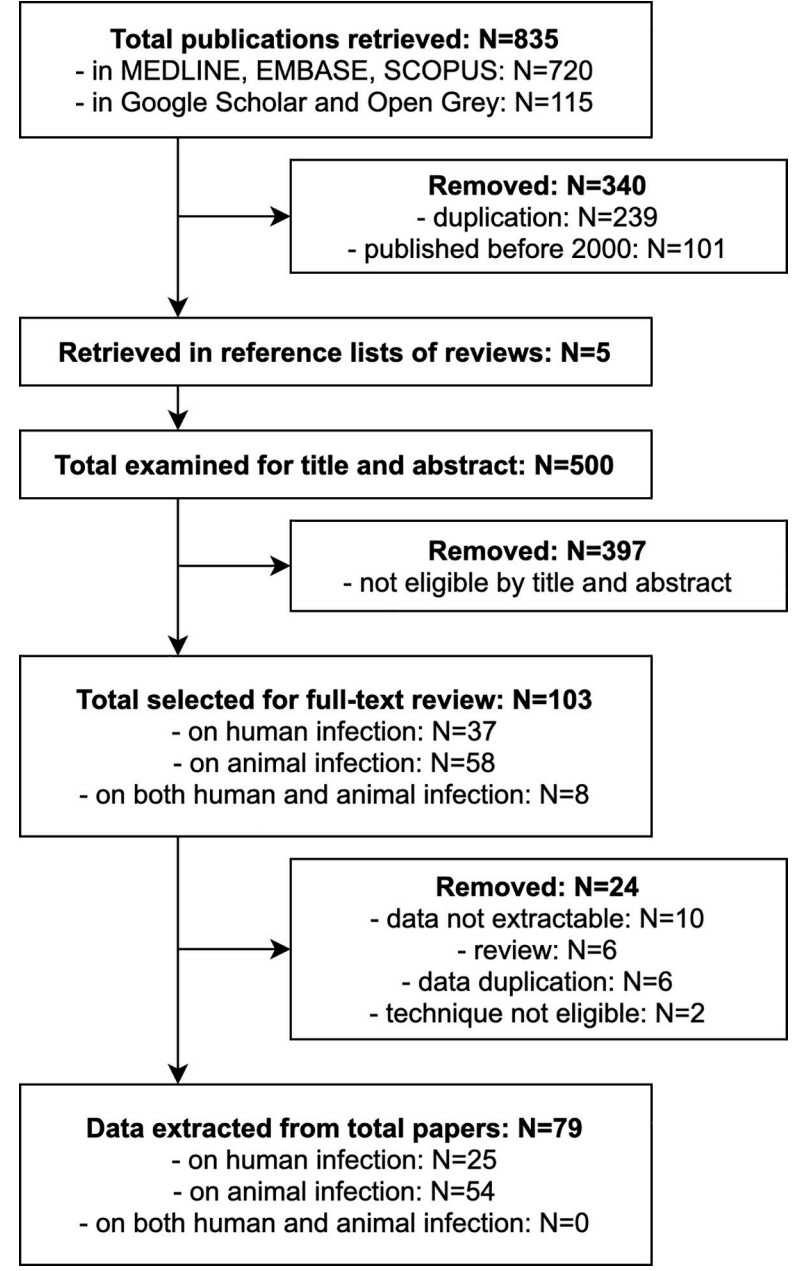

**Fig 1. Flow diagram of the literature search.**

Overall, the two papers based on abdominal ultrasound-based surveys covered 4 NUTS1 and 7 epidemiological units. The pooled prevalence and 95% CI of the two cross-sectional surveys are reported in Table 2.

## Animal infection

Among the 54 publications on animal infection, only 3 studies (all conducted in Italy) investigated the three target species (cattle, sheep, dogs) at the same time, 14 studies focused on both intermediate hosts, 9 only on cattle, 9 only on sheep and 19 papers dealt specifically with the definitive host.

**Table 1. Mean human annual incidence (n/100,000) of CE at NUTS1 level.**

| Country | NUTS1 code | N epidem. units | Years[a] | Data aggregation[b] (MY: multi-years; MA: multi-areas) | Mean annual incidence (n/100,000) | Min | Max | References |
|---|---|---|---|---|---|---|---|---|
| **Portugal** | | **1** | | | | | | |
| | PT1 | 1 | 2004–2008 | | 3.20 | | | [12] |
| **Spain** | | **124** | | | | | | |
| | ES1 | 12 | 1998–2012 | MY and MA | 0.50 | 0.00 | 1.04 | [13,14] |
| | ES2 | 27 | 1998–2012 | MY and MA | 2.71 | 0.00 | 7.38 | [13–15] |
| | ES3 | 2 | 1998–2012 | MA | 2.30 | 1.99 | 2.60 | [14] |
| | ES4 | 49 | 1996–2012 | MY and MA | 7.74 | 1.49 | 20.30 | [13,14,16–19] |
| | ES5 | 18 | 1998–2012 | MY and MA | 0.62 | 0.05 | 1.77 | [13,14] |
| | ES6 | 14 | 1998–2012 | MY and MA | 1.25 | 0.00 | 3.63 | [13,14] |
| | ES7 | 2 | 1998–2012 | MA | 0.16 | 0.12 | 0.19 | [14] |
| **France** | | **22** | | | | | | |
| | FR1 | 1 | 2005–2014 | | 0.51 | | | [20] |
| | FRB | 1 | 2005–2014 | | 0.20 | | | [20] |
| | FRC | 2 | 2005–2014 | MA | 0.47 | 0.35 | 0.59 | [20] |
| | FRD | 2 | 2005–2014 | MA | 0.21 | 0.12 | 0.29 | [20] |
| | FRE | 2 | 2005–2014 | MA | 0.19 | 0.18 | 0.19 | [20] |
| | FRF | 3 | 2005–2014 | MA | 0.54 | 0.52 | 0.57 | [20] |
| | FRG | 1 | 2005–2014 | | 0.18 | | | [20] |
| | FRH | 1 | 2005–2014 | | 0.10 | | | [20] |
| | FRI | 3 | 2005–2014 | MA | 0.24 | 0.13 | 0.40 | [20] |
| | FRJ | 2 | 2005–2014 | MA | 0.40 | 0.31 | 0.49 | [20] |
| | FRK | 2 | 2005–2014 | MA | 0.46 | 0.31 | 0.61 | [20] |
| | FRL | 1 | 2005–2014 | | 0.85 | | | [20] |
| | FRM | 1 | 2005–2014 | | 1.76 | | | [20] |
| **Italy** | | **254** | | | | | | |
| | IT0[c] | 2 | 2001–2014 | MY | 1.92 | 1.06 | 2.78 | [4] |
| | ITC | 48 | 2001–2012 | MY and MA | 0.55 | 0.00 | 1.60 | [21] |
| | ITH | 49 | 1997–2012 | MY and MA | 0.37 | 0.00 | 1.13 | [21,22] |

(*Continued*)

**Table 1.** (Continued)

| Country | NUTS1 code | N epidem. units | Years[a] | Data aggregation[b] (MY: multi-years; MA: multi-areas) | Mean annual incidence (n/100,000) | Min | Max | References |
|---|---|---|---|---|---|---|---|---|
| | ITI | 48 | 2001–2012 | MY and MA | **1.02** | 0.06 | 2.79 | [21] |
| | ITF | 72 | 2001–2012 | MY and MA | **2.76** | 0.42 | 10.80 | [21] |
| | ITG | 35 | 1998–2012 | MY and MA | **6.41** | 3.48 | 11.90 | [21,23–25] |
| **Greece** | | **2** | | | | | | |
| | EL0[d] | 2 | 1999–2000 | MY | **0.37** | 0.28 | 0.45 | [26] |
| **Romania** | | **10** | | | | | | |
| | RO1 | 1 | 2000–2010 | | **5.70** | | | [27] |
| | RO4 | 9 | 1991–2008 | MY and MA | **4.39** | 2.40 | 7.16 | [28–31] |
| **Bulgaria** | | **17** | | | | | | |
| | BG0[e] | 7 | 2006–2012 | MY | **4.85** | 3.85 | 6.28 | [32] |
| | BG3 | 9 | 2009–2013 | MY | **4.67** | 1.80 | 6.6 | [32] |
| | BG4 | 1 | 2006–2014 | | **3.58** | | | [33] |
| **Total** | | **430** | | | | | | |

NUTS1 codes explanation is provided in Supplementary Information S3 Table. NUTS1 with "0" after the two-letters initials of the country refers to the whole country.

Minimum and maximum values are reported only for NUTS1 within which more than one epidemiological unit were investigated.

[a]Years refers to the overall time frame of all included studies.

[b]MY indicates NUTS1 whose mean incidence was calculated, based on one/more studies providing data for each single year. It does not include single studies covering more years, but providing only the overall annual mean.

[c]The paper reports data for the whole country for the first (2001) and last year (2014) of the survey.

[d]The paper does not specify the reference population, but likely refers to the whole country.

[e]These data refer to the whole country.

Studies focusing on the intermediate hosts were one-year or multi-years prevalence studies generally based on slaughterhouse records of visual inspection results; only in one case, species identification was confirmed by molecular analysis. Dog populations were investigated by cross-sectional surveys in all but one study, which had a prospective longitudinal design; to make results of this study comparable with those of the other studies, only baseline data were extracted. Many different diagnostic techniques were applied, including flotation copromicroscopy or taeniid eggs isolation complemented by PCR for species identification, coproELISA, coproPCR, and necroscopic examination (variably including intestinal mucosa scraping and other poorly specified techniques).

Finally, regarding animal characteristics and subpopulations, in most cases no specific information was provided on sheep and cattle, with the exception of four studies where only adult animals (sheep >2 or 3 years; cattle >2 or 5 years) were included. Investigated dogs, instead, were usually selected according to their role or lifestyle, and the most frequent category was sheep dogs (n = 8), followed by stray or free-ranging dogs (n = 7). In a few cases also bovine farm dogs (n = 3), hunting dogs (n = 3) and pet dogs (n = 1) were included among surveyed animals.

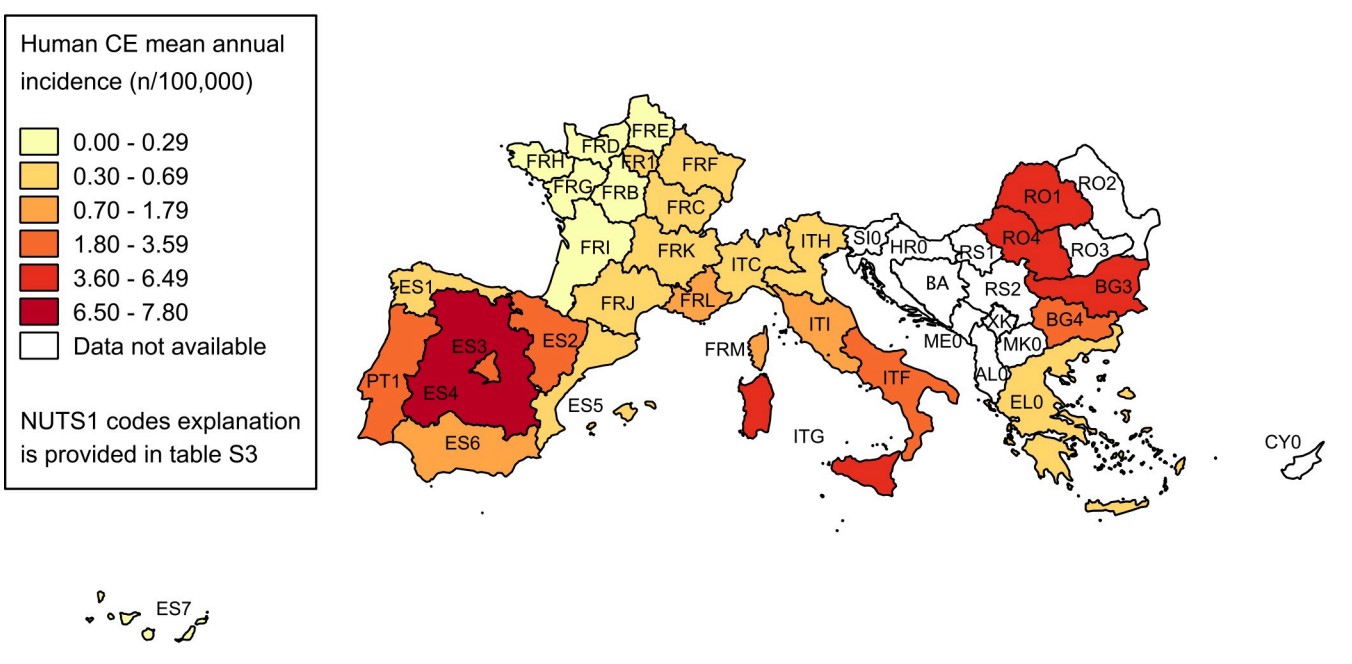

**Fig 2. Spatial distribution of average human incidence rates.** Average incidence values at NUTS1 level are visualised with progressively intense colours according to ranges established by Jenks optimization method and manually adjusted for a better results visualisation. Data at country level are reported only if more detailed data at NUTS1 level were not available (source of NUTS shapefiles: Eurostat).

Overall, extracted data were organised in 52 epidemiological units for cattle, 35 for sheep and 25 for dogs. The pooled prevalence and its 95% CI for each investigated NUTS1 are reported in Tables 3, 4 and 5, respectively for cattle, sheep and dogs. The spatial patterns of CE distributions in the three animal hosts are shown in Figs 3–5.

## Discussion

The burden of infection with *E. granulosus s.l.*, in both humans and animals, is difficult to ascertain, due to the clinical characteristics of CE in humans (a chronic, often asymptomatic or pauci-symptomatic infection, affecting mostly rural populations, with a patchy distribution

**Table 2. Human pooled prevalence at NUTS1 level.**

| Country | NUTS1 code | N epidem. units | Years[a] | Total positive humans | Total tested humans | Pooled prevalence (%) | CI95% min (%) | CI95% max (%) | References |
|---|---|---|---|---|---|---|---|---|---|
| **Romania** | | **3** | | | | | | | |
| | RO2 | 2 | 2014–2015 | 15 | 4,254 | **0.35** | 0.21 | 0.58 | [10] |
| | RO3 | 1 | 2014–2015 | 20 | 3,207 | **0.62** | 0.40 | 0.96 | [10] |
| **Bulgaria** | | **4** | | | | | | | |
| | BG3 | 2 | 2014–2015 | 10 | 3,115 | **0.32** | 0.17 | 0.60 | [10] |
| | BG4 | 2 | 2014–2015 | 23 | 6,129 | **0.35** | 0.21 | 0.58 | [10,11] |
| **Total** | | **7** | | | | | | | |

NUTS1 codes explanation is provided in Supplementary Information S3 Table.

[a]Years refers to the overall time frame of all included studies.

**Table 3. Pooled prevalence at NUTS1 level in cattle populations.**

| Country | NUTS1 code | N epidem. units | Years[a] | Total positive animals | Total tested animals | Pooled prevalence (%) | CI95% min (%) | CI95% max (%) | References |
|---|---|---|---|---|---|---|---|---|---|
| **Spain** | | **1** | | | | | | | |
| | ES2 | 1 | 2000–2006 | 683 | 40,196 | **1.70** | 1.58 | 1.83 | [34] |
| **France** | | **1** | | | | | | | |
| | FR0[b] | 1 | 2011 | 4 | 138,624 | **0.003** | 0.001 | 0.008 | [35] |
| **Italy** | | **37** | | | | | | | |
| | IT0[c] | 1 | 2009–2010 | 23 | 2,699 | **0.85** | 0.56 | 1.27 | [36] |
| | ITC | 5 | 2005–2008 | 1,473 | 731,936 | **0.22** | 0.21 | 0.23 | [37,38] |
| | ITH | 6 | 2001–2010 | 1,404 | 393,848 | **0.36** | 0.34 | 0.38 | [38–40] |
| | ITI | 6 | 1995–2010 | 9,092 | 55,358 | **16.69** | 16.30 | 17.08 | [36,38,41] |
| | ITF | 10 | 2004–2010 | 1,871 | 9,074 | **21.26** | 20.41 | 22.14 | [36,38,42–45] |
| | ITG | 9 | 2004–2015 | 3,203 | 31,182 | **42.61** | 41.46 | 43.78 | [36,38,46–51] |
| **Greece** | | **2** | | | | | | | |
| | EL5 | 1 | 2009 | 18 | 372 | **4.84** | 3.07 | 7.55 | [52] |
| | EL6 | 1 | 1999 | 2 | 792 | **0.25** | 0.06 | 1.00 | [53] |
| **Romania** | | **11** | | | | | | | |
| | RO0[d] | 4 | 2001–2012 | 164,939 | 864,390 | **19.10** | 19.02 | 19.18 | [54–57] |
| | RO1 | 2 | 2008–2011 | 3,233 | 8,009 | **40.94** | 39.81 | 40.08 | [58] |
| | RO2 | 2 | 2008–2011 | 1,269 | 5,127 | **26.04** | 24.80 | 27.33 | [58] |
| | RO3 | 1 | 2008–2011 | 1,789 | 2,791 | **64.09** | 62.30 | 65.86 | [58] |
| | RO4 | 2 | 2003–2011 | 15,750 | 87,667 | **18.12** | 17.86 | 18.38 | [58,59] |
| **Total** | | **52** | | | | | | | |

NUTS1 codes explanation is provided in Supplementary Information S3 Table. NUTS1 with "0" after the two-letters initials of the country refers to the whole country.

[a]Years refers to the overall time frame of all included studies.

[b]This study was conducted in the South of France, but it was not possible to refer data to precise NUTS2 or NUTS1.

[c]This paper did not differentiate between North-western and North-eastern Italy, reporting an aggregate value for the whole North Italy.

[d]These papers were reporting data from nation-wide surveys involving different areas, but results were presented in an aggregate way.

on the territory) and the perceived low impact on health and productivity in animals. In this systematic review, we aimed to provide a picture of the recent epidemiological situation of human and animal (cattle, sheep and dog) infection in European Mediterranean and Balkan countries, based on published literature, and to investigate the similarities in their patterns. These data may help highlight the importance of this neglected infection even in a high-resource area like Europe and define gaps in knowledge regarding its distribution. Although we did not perform a formal quality and bias assessment of the publications included, the results of our review highlight the presence of a paucity of human prevalence studies, the heterogeneity of data, and the patchy geographical coverage.

**Table 4. Pooled prevalence at NUTS1 level in sheep populations.**

| Country | NUTS1 code | N epidem. units | Years[a] | Total positive animals | Total tested animals | Pooled prevalence (%) | IC95% min (%) | IC95% max (%) | References |
|---|---|---|---|---|---|---|---|---|---|
| **Spain** | | **1** | | | | | | | |
| | ES2 | 1 | 2000–2006 | 88 | 88,369 | **0.10** | 0.08 | 0.12 | [34] |
| **France** | | **1** | | | | | | | |
| | FR0[b] | 1 | 2010 | 27 | 725,903 | **0.004** | 0.003 | 0.005 | [35] |
| **Italy** | | **13** | | | | | | | |
| | ITC | 1 | 2006 | 3 | 822 | **0.36** | 0.11 | 1.12 | [37] |
| | ITI | 2 | 1995–2004 | 14,829 | 25,225 | **58.09** | 57.46 | 58.72 | [41,60] |
| | ITF | 1 | 2006 | 77 | 365 | **21.09** | 16.90 | 25.30 | [45] |
| | ITG | 9 | 1995–2010 | 4,821 | 6,998 | **68.50** | 67.38 | 69.59 | [46–48,61–65] |
| **Greece** | | **9** | | | | | | | |
| | EL5 | 4 | 2009–2015 | 336 | 1,140 | **29.53** | 26.95 | 32.25 | [66] |
| | EL6 | 5 | 1999–2015 | 1,080 | 6,906 | **30.08** | 28.54 | 31.68 | [53,63,66–68] |
| **Romania** | | **11** | | | | | | | |
| | RO0[c] | 4 | 2001–2012 | 78,421 | 600,829 | **13.16** | 13.07 | 13.25 | [54–57] |
| | RO1 | 2 | 2008–2011 | 2,942 | 5,834 | **50.28** | 48.99 | 51.58 | [58] |
| | RO2 | 2 | 2008–2011 | 5,672 | 12,443 | **45.70** | 44.81 | 46.58 | [58] |
| | RO3 | 1 | 2008–2011 | 1,341 | 1,951 | **68.73** | 66.64 | 70.75 | [58] |
| | RO4 | 2 | 2003–2011 | 8,690 | 83,942 | **11.07** | 10.84 | 11.30 | [58,59] |
| **Total** | | **35** | | | | | | | |

NUTS1 codes explanation is provided in Supplementary Information S3 Table. NUTS1 with "0" after the two-letters initials of the country refers to the whole country.

[a]Years refers to the overall time frame of all included studies.

[b]The study was conducted in the South of France, but it was not possible to refer data to precise NUTS2 or NUTS1.

[c]These papers were reporting data from nation-wide surveys involving different areas, but results were presented in an aggregate way.

Results obtained from the analysis of human-based studies on CE highlight the paucity of prevalence studies, the heterogeneity of incidence data that could be retrieved for the analysis (average annual incidence rates at NUTS1 level ranged from 0.10 to 7.74 per 100,000), and the patchy geographical coverage of the studies, with lack of such data especially for the Balkans. As mentioned, CE is a neglected, chronic infection often asymptomatic or pauci-symptomatic for a long time or even indefinitely. The population of individuals with CE can be schematically described as a pyramid composed of: i) the "tip" formed by cases reaching medical attention and hospitalized, therefore included in official HDR-based statistics; ii) a "stratum" of cases reaching medical attention but not requiring hospitalization and therefore only variably captured in medical records; and iii) the "basis" formed by the likely large proportion of cases never reaching medical attention. This last portion of individuals with CE can only be quantified using population-based screening campaigns, which are extremely scant in the retrieved published literature. Consequently, prevalence and clinical-based studies are complementary and not mutually exclusive, not only for the understanding of CE epidemiology on a territory,

**Table 5. Pooled prevalence at NUTS1 level in dog populations.**

| Country | NUTS1 code | N epidem. units | Years[a] | Total positive animals | Total tested animals | Pooled prevalence (%) | IC95% min (%) | IC95% max (%) | References |
|---|---|---|---|---|---|---|---|---|---|
| **Portugal** | | 4 | | | | | | | |
| | PT1 | 4 | 2009–2016 | 1 | 1,086 | **0.95** | 0.13 | 6.45 | [69–72] |
| **Spain** | | 4 | | | | | | | |
| | ES2 | 2 | 1998–2003 | 106 | 1,761 | **11.86** | 9.89 | 14.18 | [73,74] |
| | ES6 | 2 | 2004–2005 | 0 | 350 | **0.00** | 0.00 | 0.84[b] | [75,76] |
| **France** | | 1 | | | | | | | |
| | FMR | 1 | 2013–2014 | 3 | 259 | **1.16** | 0.37 | 3.53 | [77] |
| **Italy** | | 7 | | | | | | | |
| | ITC | 2 | 2003–2006 | 19 | 66 | **28.83** | 19.20 | 40.85 | [37,78] |
| | ITH | 1 | 2012–2017 | 2 | 208 | **0.96** | 0.24 | 3.76 | [79] |
| | ITI | 1 | 2003 | 5 | 106 | **4.72** | 1.98 | 10.83 | [78] |
| | ITF | 1 | 2006 | 36 | 113 | **31.86** | 23.94 | 40.97 | [45] |
| | ITG | 2 | 2003–2005 | 40 | 352 | **23.17** | 16.47 | 31.57 | [46,80] |
| **Albania** | | 1 | | | | | | | |
| | AL0 | 1 | 2009 | 3 | 111 | **2.70** | 0.87 | 8.04 | [81] |
| **Kosovo** | | 2 | | | | | | | |
| | Kosovo | 2 | 2004–2013 | 10 | 809 | **1.53** | 0.82 | 2.82 | [82,83] |
| **Romania** | | 3 | | | | | | | |
| | RO1 | 1 | 2008 | 364 | 1,892 | **19.24** | 17.52 | 21.08 | [84] |
| | RO3 | 1 | 2011 | 8 | 86 | **9.30** | 4.72 | 17.51 | [85] |
| | RO4 | 1 | 2011 | 4 | 46 | **8.67** | 3.30 | 20.97 | [85] |
| **Bulgaria** | | 2 | | | | | | | |
| | BG3 | 1 | 2014 | 0 | 40 | **0.00** | 0.00 | 7.33[b] | [86] |
| | BG4 | 1 | 2014 | 0 | 40 | **0.00** | 0.00 | 7.33[b] | [86] |
| **Cyprus** | | 1 | | | | | | | |
| | CY0 | 1 | 2001 | 184 | 6,489 | **2.84** | 2.46 | 3.27 | [87] |
| **Total** | | 25 | | | | | | | |

NUTS1 codes explanation is provided in Supplementary Information S3 Table. NUTS1 with "0" after the two-letters initials of the country refers to the whole country.

[a]Years refers to the overall time frame of all included studies.

[b]When no animal resulted positive, the maximum expected prevalence value was calculated theoretically, considering a population N = 10,000 and the overall sampling size n (i.e. n = 350 in ES6 and n = 40 in BG3 and BG4).

but also for burden of disease estimates and public health evaluations. Indeed, clinically relevant cases (i.e. where the infection has a health impact requiring treatment) can be present in all above-mentioned categories [10,88], due to the proteiform clinical characteristics of CE and its neglected status, with generally poor awareness of both at-risk populations and health care personnel, and consequent misdiagnosis and underreporting. Concerning clinical-based studies, major limitations of most studies for the understanding of CE epidemiology are their heterogeneity in terms of type of cases included (such as all cases reaching medical attention,

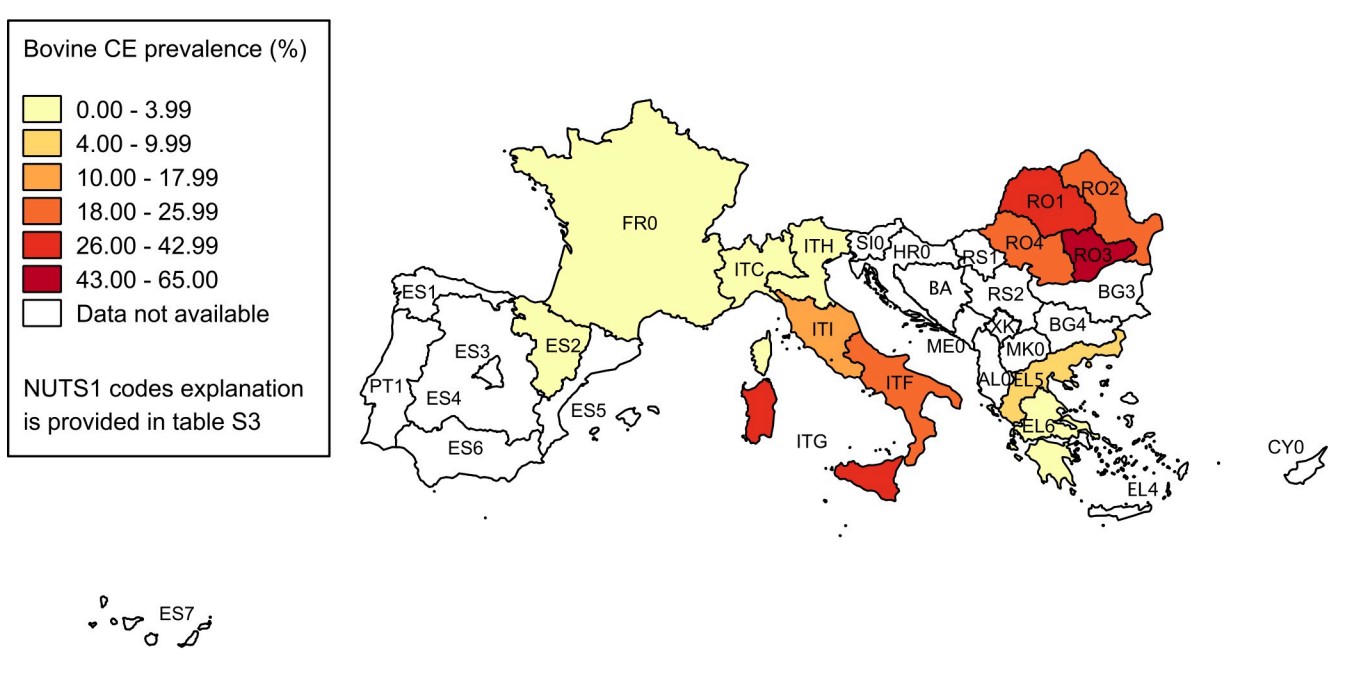

**Fig 3. Spatial distribution of CE prevalence in cattle.** Pooled prevalence values at NUTS1 level are visualised with progressively intense colours according to ranges established by Jenks optimization method and manually adjusted for a better results visualisation. Data at country level are reported only if more detailed data at NUTS1 level were not available (source of NUTS shapefiles: Eurostat).

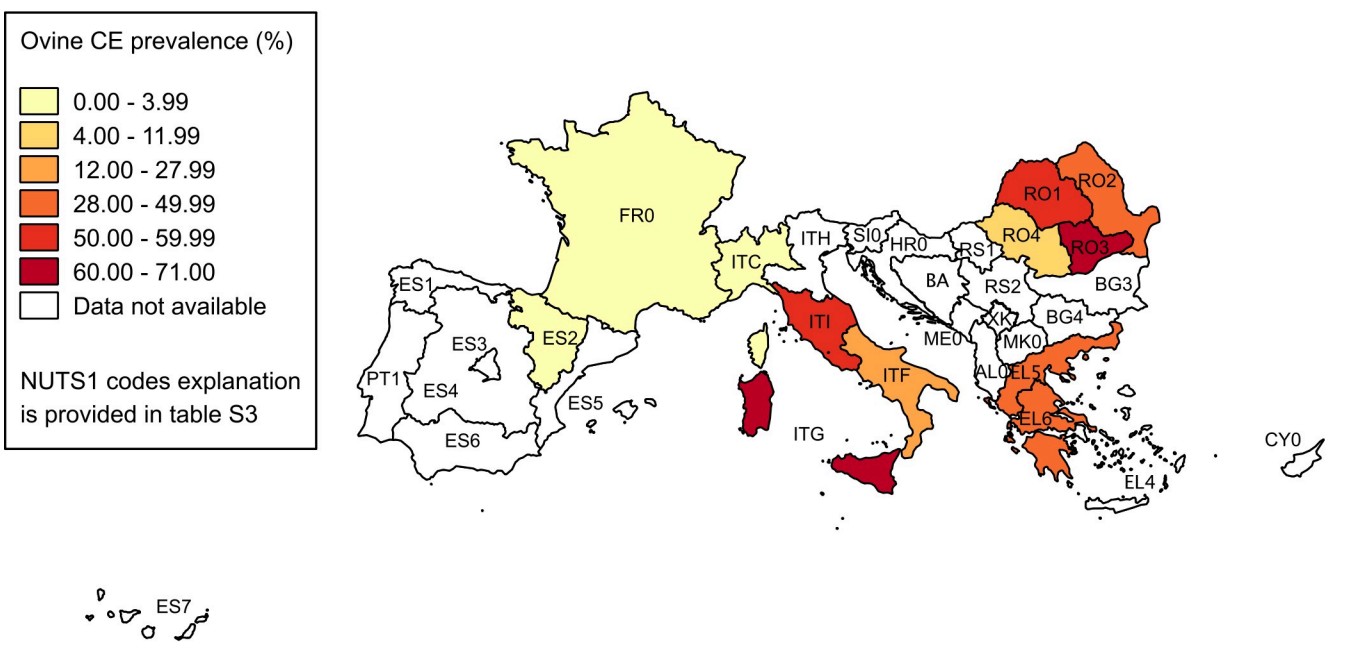

**Fig 4. Spatial distribution of CE prevalence in sheep.** Pooled prevalence values at NUTS1 level are visualised with progressively intense colours according to ranges established by Jenks optimization method and manually adjusted for a better results visualisation. Data at country level are reported only if more detailed data at NUTS1 level were not available (source of NUTS shapefiles: Eurostat).

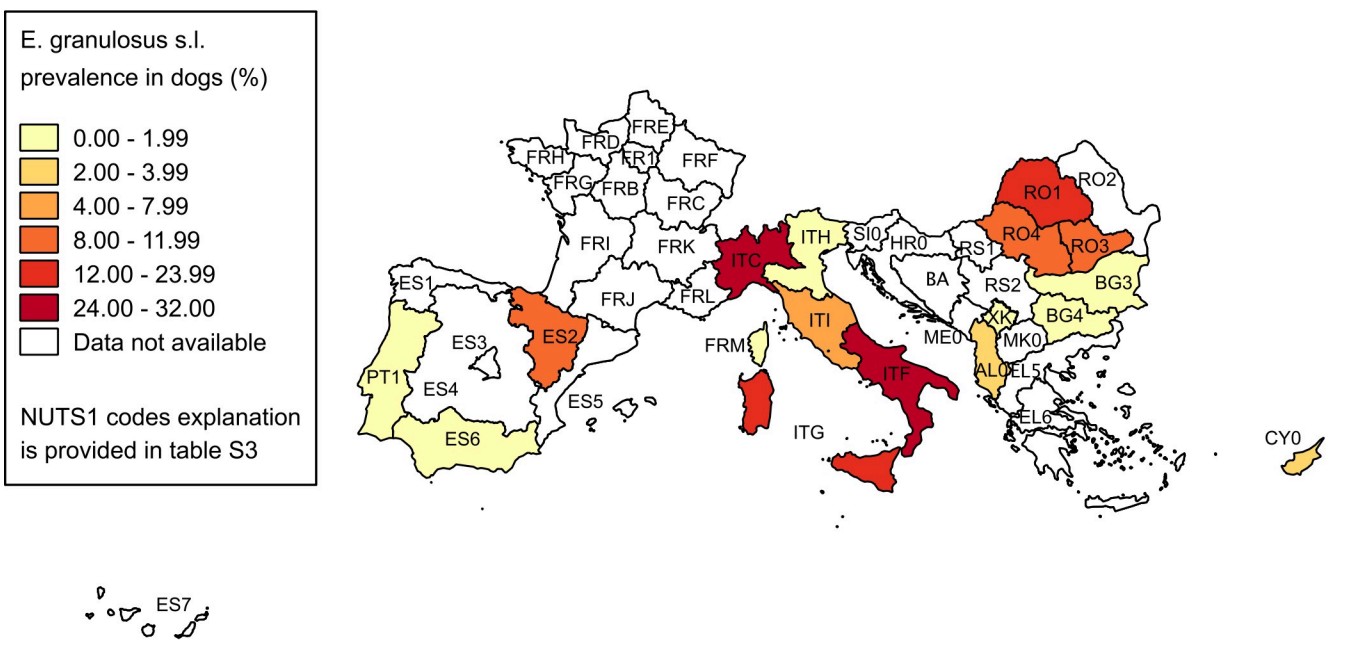

**Fig 5. Spatial distribution of *E. granulosus s.l.* prevalence in dogs.** Pooled prevalence values at NUTS1 level are visualised with progressively intense colours according to ranges established by Jenks optimization method and manually adjusted for a better results visualisation. Data at country level are reported only if more detailed data at NUTS1 level were not available (source of NUTS shapefiles: Eurostat).

or hospitalized cases, or surgically confirmed ones, or only residents in a particular area) and the frequent lack of reporting of the at-risk population to which the incidence estimate refers. Furthermore, being CE a chronic and complex infection requiring long-term follow-up, distinction between incidence of first clinical diagnoses and incidence simply calculated using retrospective clinical records would be important but was seldom specified. In this light, at least to capture all cases reaching medical attention, it would be pivotal to implement a compulsory notification system, including both hospitalized patients and outpatients. The European Register of CE (ERCE) [89], if adopted at national level by health authorities, could represent a suitable platform for such a system.

For what concerns the spatial distribution of human CE, the results of our study, granted the absence of data from many Balkan countries, indicate Southern (ITF) and insular (ITG) Italy, Spain, and Eastern Europe (Bulgaria and Romania) as the most affected areas, in line with other recent narrative reviews [1] and official WHO data [90]. Minor differences from these figures are probably due to the different criteria used for data inclusion. The large differences between minimum and maximum incidence estimate calculated for some NUTS1 derive either from differences over time for the same area (e.g. Stara Zagora region in Bulgaria–BG3) or from both differences in the values of included NUTS2 and across the investigated period, such as in Southern and insular Italy, and in Central Spain (ES4). Also, the different inclusion criteria for cases series (all clinically observed cases or hospitalized cases) and the frequently unclear definition of the population at risk had an important impact on the estimation of incidence rate, as evidenced by the studies in North-eastern (ES2) and Central Spain [13–15,17].

Also the data on animal infection are not uniformly distributed in the study area, as demonstrated by the differences in number of epidemiological units investigated in selected countries, with Italy and Romania generally representing the most investigated countries. As for

human studies, data for cattle and sheep were not available for the Balkan area. Besides, few and spatially limited studies on animal hosts were retrieved for the Iberian Peninsula.

Studies on intermediate hosts showed a good level of homogeneity in methodology, being all based on retrospective slaughterhouse surveys, but had ample variability in results. In all investigated populations, at least one positive animal was found, and pooled prevalence values at NUTS1 level varied from 0.003% to 64.09% in cattle, and from 0.004% to 68.73% in sheep. All animals slaughtered in a given area/abattoir were generally considered as the population at risk, and only in a few studies young animals were excluded. Most of the studies were based on the passive surveillance activity (meat inspection) routinely implemented in slaughterhouses, thus most likely ensuring sufficiently similar diagnostic performances among all investigations. This aspect suggests that data extracted from these papers are comparable for the whole study area. Since the population at risk (denominator) was reported in all studies, we were able to calculate the pooled prevalence for each NUTS1. This provided a weighted value, accounting for individual study weights due to different sample sizes, in the investigations conducted in the same epidemiological unit. Higher prevalence values were similarly found for sheep and cattle in Central (ITI), Southern and insular Italy, and in Romania (>10%), whereas high rates were encountered only for sheep in Greece. Northern Italy (ITC and ITH) and France reported prevalence values below 1%, confirming their status of hypo-endemic areas [1].

Dog populations were mostly investigated through cross-sectional surveys, showing mean prevalence values ranging from 0 to 31.86%. The differences in prevalence rates reflect the variability among areas, and the distribution pattern only partly resembles that of human incidence (Figs 1 and 4). However, these differences can be due also to the use of different investigation approaches, both in terms of diagnostic techniques and specific target population. Most studies targeted at-risk dogs (i.e. with higher probability of accessing raw infected offal of intermediate hosts) such as sheep dogs or free-ranging dogs, since their main objective was to find positive dogs. Therefore, in consideration of the general heterogeneity of inclusion criteria of dogs among these investigations, the comparison of results is difficult. Besides, the ample variability of diagnostic techniques used by different studies is alone sufficient to hamper any possibility of comparing data among areas. In some cases, studies using coproELISA techniques showed an unexpectedly high prevalence (e.g. North-western Italy—ITC), suggesting a possible cross-reactivity problem, as already described for this technique and generally occurring when infection with other Taenidae, particularly *Taenia hydatigena*, is common in the area [91].

In the present study we investigated differences in spatial distribution of *E. granulosus s.l.*, referring data extracted from the available literature to an artificial epidemiological unit (one NUTS2 in one year) created *ad hoc*. The intention was to make extracted data as much homogeneous as possible in terms of investigated population, allowing a fair comparison among areas. Actually, data obtained from literature were extremely variable from different points of view, including the real extension of studied populations (e.g. only a limited area/population of the NUTS2 was investigated in some publications). In the case of cross-sectional studies, we addressed the problem of limited homogeneity among studies in animals (all papers) and humans (only 2 papers), calculating the pooled prevalence, which accounts for the differences in the investigated populations size (i.e. sampling fraction). Coming to retrospective case series studies in humans reporting incidence data (all but two papers), it should be noted that these studies usually considered all the inhabitants of the area under investigation as the population at risk, which therefore generally consisted of hundreds thousands or millions individuals. In our review, the single-year figure for human population at risk could be estimated only in the 87 epidemiological units of the 4 papers where incidence value was associated with the number of new cases for each year. In these units the population at risk ranged from a minimum of

73,529 to a maximum of 6,250,000 persons, but it was comprised in most cases (75%; 66/87) between 500,000 and 3,000,000.

In consideration of the chronic nature of CE in humans, we decided that it was of little value to attempt the identification of overall temporal trends in the time frame fixed for our literature search. Coming to infection in animal hosts, few areas were investigated for more than two consecutive years. As a consequence, we did not compare incidence and prevalence data among different periods in the years (1995–2019) covered by our review. However, it is worth noting that some multi-year studies described such kind of temporal trends, usually showing a decreasing trend of the disease burden in humans [14,17,21,32], whereas in animals both decreasing and stable trends were reported [34,41,65].

The inhomogeneous geographical coverage of human and animal studies prevented a formal analysis of the relation between human and animal infection epidemiology. However, the geographical distribution of *E. granulosus s.l.* infection in different territories of European Mediterranean and Balkan countries presents high similarities between human and animal hosts, as shown by the four maps. In particular, both bovine and ovine prevalence patterns parallel quite well human incidence distribution, at least for the NUTS1 where data are available for all species. In our literature search, few papers reported data from both humans and animals, and only three investigated the correlation between human and animal data, reporting that the reduction in infection rates in dogs and livestock corresponded to a similar decline in human CE incidence [13,15,26]. Our analysis did not allow for a statistically based inference on the relation between prevalence data in animal hosts and human incidence, but results from NUTS investigated for both human and animal infection suggest that a relation can be assumed. It is well known that in Mediterranean and Balkan countries the transhumant sheep farming is considered as the main driver for *E. granulosus s.l.* maintenance [92,93]. However, in our results, bovine data showed a more similar pattern to human data compared to sheep, suggesting its potential role as sentinel species for estimating the risk for humans to get infected with CE, in areas where both livestock breeding is practiced, as previously hypothesized [43,79]. This could be in part due to the more similar condition of the human and bovine host in respect to CE and its diagnosis, compared to sheep, i.e. a longer average lifespan of cattle allows for a more extended time from infection up to the moment when this is evidenced at slaughterhouse. Since *E. granulosus sensu strictu* G1-G3 are the genotypes mainly affecting bovines in Europe [94], the higher similarities in epidemiological features of infection between cattle and humans may be also due to their common role of accidental intermediate hosts for these genotypes. A quantitative description of circulating genotypes could help in a better interpretation of the relation between human and animal prevalence data. This information is available at global level [95] or more specifically in other geographical context [96], but not for the areas included in our review. A more detailed description was not attempted in this study, because of the paucity of relevant data in the included papers on animal infection and the absence of such information from studies on humans.

It is worth mentioning that in this review we did not include data of livestock intermediate hosts other than sheep and cattle that can have a role in the human infection epidemiology, such as swine and goats. This decision was based on the observation that other hosts seem of lower relevance for the epidemiology of human CE in the investigated geographical area [1, 94]. Equally, we did not address *E. granulosus s.l.* infection in wild definitive and intermediate hosts such as sylvatic canids and wild boars or wild ungulates. However, the investigation of the epidemiology of CE in other animals, especially livestock, is surely of interest and should be envisaged as a future, complementary systematic review.

Finally, the results of our analysis highlight the notorious problem of underreporting of CE in humans. As an example, if we consider the recent work by Piseddu and colleagues

examining HDR for CE in Italy from 2001 to 2014 [4], the median of CE hospitalizations per year was 848. This figure alone roughly equals all cases of echinococcosis (both cystic and alveolar echinococcosis) reported by all Member States at European level yearly [6] but Italy does not even report HDR data at European level. Substantial underreporting is also indicated by data from other countries; for example in 2013 in Romania only 55 CE cases were reported in official European statistics for the whole country [97], while one hospital alone in the capital, for the same year, recorded 104 CE cases [10].

Also data on animal infection extracted from the reviewed literature showed evident differences with official European Union yearly reports, where a specific table on number of animals positive/tested for *E. granulosus s.l.* has been included since 2015 [6,98–100]. As an example, in these reports, the prevalence reported for sheep in Greece was constantly <2%, whereas values near to 30% were found in our review. Similarly, values generally <5% are officially reported for Italian sheep, whereas half of the Italian NUTS included in the present study showed pooled prevalence higher than 50%.

In conclusion, this systematic review provides a rigorous summary of the epidemiological situation of *E. granulosus s.l.* infection in humans and in selected domestic animals, based on published literature, demonstrating that prevalence values for bovine intermediate host parallel quite well human incidence. While studies on animal intermediate hosts were sufficiently well defined and homogeneous, studies on humans were extremely heterogeneous and lacking important information, such as the precise definition of the population at risk, inclusion criteria for cases, and sometimes also the number of newly diagnosed cases. Also, prevalence studies were extremely scant. The implementation of new studies on human incidence and prevalence and bovine prevalence, appropriately defined in their methodology and covering the geographical areas still not investigated, could help in completing the picture already drafted by our review. The results of our study show the importance of this neglected infection in the study area and strongly prompt public health authorities to implement surveillance strategies for both human and animal infection.

## Supporting information

**S1 PRISMA Check list.**
(DOCX)

**S1 Text. Search strategy.**
(DOCX)

**S1 Table. Human CE Dataset.**
(XLSX)

**S2 Table. Animal CE Dataset.**
(XLSX)

**S3 Table. List of NUTS1 codes with relative extended names.**
(XLSX)

## Author Contributions

**Conceptualization:** Francesca Tamarozzi, Rudi Cassini.

**Data curation:** Francesca Tamarozzi, Matteo Legnardi, Andrea Fittipaldo, Michele Drigo, Rudi Cassini.

**Formal analysis:** Michele Drigo.

**Funding acquisition:** Rudi Cassini.

**Investigation:** Francesca Tamarozzi, Matteo Legnardi, Michele Drigo.

**Project administration:** Rudi Cassini.

**Resources:** Andrea Fittipaldo.

**Visualization:** Matteo Legnardi.

**Writing – original draft:** Francesca Tamarozzi, Rudi Cassini.

**Writing – review & editing:** Francesca Tamarozzi, Matteo Legnardi, Andrea Fittipaldo, Michele Drigo, Rudi Cassini.

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
