## [Decision Letter · Decision Letter 0]

19 May 2020

Dear Prof. Cassini,

Thank you very much for submitting your manuscript "Epidemiological distribution of human and animal cystic echinococcosis in European Mediterranean and Balkan countries: a Systematic Review." for consideration at PLOS Neglected Tropical Diseases. As with all papers reviewed by the journal, your manuscript was reviewed by members of the editorial board and by several independent reviewers. In light of the reviews (below this email), we would like to invite the resubmission of a significantly-revised version that takes into account the reviewers' comments. 

We cannot make any decision about publication until we have seen the revised manuscript and your response to the reviewers' comments. Your revised manuscript is also likely to be sent to reviewers for further evaluation.

Sincerely,

Paul Robert Torgerson

Guest Editor

Adriano Casulli

Deputy Editor

Reviewer's Responses to Questions

**Key Review Criteria Required for Acceptance?**

**Methods**

-Are the objectives of the study clearly articulated with a clear testable hypothesis stated?

-Is the study design appropriate to address the stated objectives?

-Is the population clearly described and appropriate for the hypothesis being tested?

-Is the sample size sufficient to ensure adequate power to address the hypothesis being tested?

-Were correct statistical analysis used to support conclusions?

-Are there concerns about ethical or regulatory requirements being met?

Reviewer #1: The objective of the study are clearly defined, the study design is appropriate, the sample size (papers for a systematic review) sufficient to address the hypothesis being tested, and the statistical analysis is correct. There are no concerns on ethical and regulatory requirements.

Reviewer #2: Objectives are clearly articulated. Study design is appropriate and statistics are correct for the analysis performed

Reviewer #3: The first sentence of the Materials and Methods refers to “the research question”. However, a research question is not explicitly stated. 

Please indicate why the specific databases (and web search engines) were chosen and why the original search dates differed between sites. It is also not clear what #20-#24 refer to in the Embase search strategy (S1 text). 

Please elaborate on why only the first 500 search results were considered for the Google Scholar, Scopus, and Open Grey searches? Where these sorted by relevance or date? 

What is meant by “publication status” (line 111)?

It appears that data could be included from 1980 onward, but only articles published after 2000 were included in the review. How do the authors justify the 1980 data cutoff (line 115)?

The authors need to provide an overview of the NUTS system and elaborate on what constitutes the NUTS1 and NUTS2 levels. Without this information in the main text, the paper becomes very difficult to follow. 

I would suggest that the NUTS1 codes be included in the actual paper (or the NUTS1 codes swapped for the region names in the tables).

**Results**

-Does the analysis presented match the analysis plan?

-Are the results clearly and completely presented?

-Are the figures (Tables, Images) of sufficient quality for clarity?

Reviewer #1: The analysis presented match the analysis plan. The results are clearly presented including tables and maps of good quality for clarity.

Reviewer #2: Results are clearly and completely presented. Figures and tables are clear and well presented

Reviewer #3: Since data appear to be included from 1980 onward, it would be helpful if the tables provided the time frames encompassed by the included studies.

It appears that, in many publications, disease frequency (e.g., incidence) was calculated by the original authors. Did the review authors use any method to assess level of bias in the included articles?

Figure 5- I assume the authors mean E. granulosus s.l. prevalence and not CE prevalence since dogs are infected with the adult form of the parasite. 

Figures- prevalence and incidence bins are not mutually exclusive (e.g., would an area where the prevalence in cattle is 4.0% go in the 0.0-4.0 bin or the 4.0-10.0 bin).

**Conclusions**

-Are the conclusions supported by the data presented?

-Are the limitations of analysis clearly described?

-Do the authors discuss how these data can be helpful to advance our understanding of the topic under study?

-Is public health relevance addressed?

Reviewer #1: The conclusions are supported by the data presented. However, the limitation of the analysis is not clearly described. The topic is of public health importance and the public health relevance is well addressed.

Reviewer #2: See summary and general comments

Reviewer #3: The authors discuss how differences in the clinical presentation of CE patients (e.g., surgical patients versus patients being treated medically or detected incidentally) impact the available data. That being said, it would be helpful if data type was included in table 1. Along those same lines, inclusion of diagnostic technique would assist with understanding the values produced for definitive hosts (table 5)?

Again, please clarify the time frame for the data included in the review (line 432).

The authors spend quite a bit of the Discussion trying to explain relationships that they earlier state could not be evaluated (e.g., in support of the authors’ previously published hypothesis regarding the use of cattle as sentinel species). Therefore, they may want to temper the language somewhat.

**Editorial and Data Presentation Modifications?**

Reviewer #1: Minor revision

Reviewer #2: (No Response)

Reviewer #3: Title: It appears that the authors not only evaluated cystic echinococcosis (i.e., infection with the larval form of the parasite), but also adult parasite infection in definitive hosts. This is not reflected in the title. 

Abstract: NUTS1, NUTS2, and epidemiological unit are not well defined in the abstract. A clearer link needs to be made between the presented principal findings and the conclusion that Italy, Spain, and Eastern Europe are the most affected areas.

**Summary and General Comments**

Reviewer #1: The manuscript PNTD-D-20-00593 by Tamarozzi et al. titled “Epidemiological distribution of human and animal cystic echinococcosis in European Mediterranean and Balkan coutries: a Systematic Rewier” describes the results obtained from a systematic literature review on Echinococcus granulosus in humans and animals. The aim of this study is to give an overview on the distribution and epidemiology of cystic echinococcosis in Europe, and specifically in European Mediterranean and Balkan countries, where this disease is endemic. For this purpose, the authors considered all possible studies that could help to answer the research question. Thus, cross-sectional and longitudinal studies, as well as case series, reporting prevalence or incidence of human or animal (ovine, bovine, and canine) E. granulosus s.l. infection were included. To ensure maximum data accuracy, only papers/studies reporting the species of Echinococcus investigated, the number of people/animals surveyed, the number of positive cases, details about the methodology of diagnosis and the geographical site were included. Each data obtained, separated for host species, was then referred to a specific NUTS (Nomenclature of Territorial Units for Statistics). The work is presented according to the recommendations of the Preferred Reporting Items for Systematic Reviews and Meta-Analyses.

The topic is of interest due to the impact of cystic echinococcosis (CE) on human and animal health, but also for the economic losses associated. There is a growing interest on CE and a comprehensive review of its distribution in definitive and intermediate hosts in endemic areas, as those described in the work, could improve the targeting of control measures and to enhance the cost-effectiveness of integrated disease control programmes for E. granulosus.

Thus, I recommend the publication of the paper in PLOS Neglected Tropical Diseases after minor revisions. The paper is well written, the questions posed by the authors are well defined, the methods appropriate, and the results, discussion and conclusions well balanced and adequately supported by the data.

GENERAL COMMENTS

- Authors should explain why they didn’t consider animal intermediate hosts other than cattle and sheep, e.g. swine and equids. Furthermore, the role of sylvatic canids and wild intermediate hosts (e.g. wild boar and wild ungulates) should be considered in the epidemiology of E. granulosus.

- I was wondering how the authors have treated farm prevalence data in their analysis.

- I would suggest the authors to add data on genotypes of E. granulosus if available from the studies used for their review.

- The Authors often mention the limitation of the studies used in their review. But wat about the limitation of their approach?

- From a practical point of view, I would suggest to emphasize how the maps and the outcome of the review will be transferred to policy makers and stake holders, in order to improve surveillance of this important neglected disease as well as in order to plan effective control measures against E. granulosus

SPECIFIC COMMENTS

- Line 94: At the end of the introduction, the authors explain the aim of this work, specifying that the systematic review was carried out between 1980 and 2019, whereas in the abstract they mention “between 2000 and 2019”. Moreover, since papers before 2000 were not considered in the review, I would suggest to change to “between 2000 and 2019” throughout the study.

- Line 107: Please change “Echinococus” to “Echinococcus”. 

- Line 117, Please provide a reference for PRISMA

Reviewer #2: In general, this is a very interesting review on the status of CE in human and animla hosts, and gives a good idea on the underrreporting of official data. The highlights of this systematic review (underreporting, centinel animals and geograpical distribution) are of importance not only for the scientific community, but also for the policy makers.

Minor comments: Authors should doublé-check if articles on animal CE refer to animals only slaughtered in a given geographical area or breeded/fed in the geographical area or reference for each article. Slaughterhouses in EU are now big and are usually "specialized" in a specific animal species, and usually receive animals of that species from all geographical regions inside the country. Geographical origin (and not place of slaughter) of the CE infected animals should be cross-checked.

A second question of concern is the age of animals. Sheep reaching slaughetrhouses and subjected to inspection are in many cases very young animals. Maybe this could explain the best matching between human-cow numbers and not between human-sheep numbers. This should be commented in the text

Reviewer #3: This is a systematic review looking at Echinococcus granulosus senu lato infection in humans and animal hosts in Mediterranean and Balkan countries. The authors use the Nomenclature of Territorial Units for Statistics (NUTS) to analyze and present their findings. However, the NUTS system (and the reason for using this system) is not well described, making it difficult for the reader to interpret what the findings mean from a country level and control perspective.

PLOS authors have the option to publish the peer review history of their article (what does this mean?). If published, this will include your full peer review and any attached files.

Reviewer #1: No

Reviewer #2: No

Reviewer #3: No
---

## [Editor Report · Decision Letter 1]

18 Jun 2020

Dear Prof. Cassini,

Thank you very much for submitting your manuscript "Epidemiological distribution of Echinococcus granulosus s.l. infection in human and domestic animal hosts in European Mediterranean and Balkan countries: a Systematic Review" for consideration at PLOS Neglected Tropical Diseases. As with all papers reviewed by the journal, your manuscript was reviewed by members of the editorial board and by several independent reviewers. The reviewers appreciated the attention to an important topic. Based on the reviews, we are likely to accept this manuscript for publication, providing that you modify the manuscript according to the review recommendations. 

There is a minor suggestion. For human incidence, authors should always use "annual incidence per 100,000" rather than "incidence per 100,000". This is to standardize disease reporting and to ensure the incidence is reported as annual incidence and not over some other period of time.

Sincerely,

Paul Robert Torgerson

Guest Editor

Adriano Casulli

Deputy Editor

There is a minor suggestion. For human incidence, authors should always use "annual incidence per 100,000" rather than "incidence per 100,000". This is to standardize disease reporting and to ensure the incidence is reported as annual incidence and not over some other period of time
---

## [Editor Report · Decision Letter 2]

25 Jun 2020

Dear Prof. Cassini,

We are pleased to inform you that your manuscript 'Epidemiological distribution of Echinococcus granulosus s.l. infection in human and domestic animal hosts in European Mediterranean and Balkan countries: a Systematic Review' has been provisionally accepted for publication in PLOS Neglected Tropical Diseases.

Best regards,

Paul Robert Torgerson

Guest Editor

Adriano Casulli

Deputy Editor

---

## [Editor Report · Acceptance letter]

3 Aug 2020

Dear Prof. Cassini,

We are delighted to inform you that your manuscript, "Epidemiological distribution of Echinococcus granulosus s.l. infection in human and domestic animal hosts in European Mediterranean and Balkan countries: a Systematic Review," has been formally accepted for publication in PLOS Neglected Tropical Diseases.

Best regards,

Shaden Kamhawi

co-Editor-in-Chief

Paul Brindley

co-Editor-in-Chief
